

# The genome of the butternut canker pathogen, *Ophiognomonia clavigignenti-juglandacearum* shows an elevated number of genes associated with secondary metabolism and protection from host resistance responses

Guangxi Wu[1], Taruna A. Schuelke[2], Gloria Iriarte[3] and Kirk Broders[3]

[1] Department of Agricultural Biology, Colorado State University, Fort Collins, CO, USA
[2] Ecology, Evolution and Marine Biology Department, University of California, Santa Barbara, Santa Barbara, CA, USA
[3] Smithsonian Tropical Research Institute, Balboa, Ancon, Republic of Panama

Corresponding author
Kirk Broders, brodersk@si.edu

## ABSTRACT

*Ophiognomonia clavigignenti-juglandacearum* (*Oc-j*) is a plant pathogenic fungus that causes canker and branch dieback diseases in the hardwood tree butternut, *Juglans cinerea*. *Oc-j* is a member of the order of Diaporthales, which includes many other plant pathogenic species, several of which also infect hardwood tree species. In this study, we sequenced the genome of *Oc-j* and achieved a high-quality assembly and delineated its phylogeny within the Diaporthales order using a genome-wide multi-gene approach. We also further examined multiple gene families that might be involved in plant pathogenicity and degradation of complex biomass, which are relevant to a pathogenic life-style in a tree host. We found that the *Oc-j* genome contains a greater number of genes in these gene families compared to other species in the Diaporthales. These gene families include secreted CAZymes, kinases, cytochrome P450, efflux pumps, and secondary metabolism gene clusters. The large numbers of these genes provide *Oc-j* with an arsenal to cope with the specific ecological niche as a pathogen of the butternut tree.

## INTRODUCTION

*Ophiognomonia clavigignenti-juglandacearum* (*Oc-j*) is an Ascomycetous fungus in the family Gnomoniaceae and order Diaporthales. Like many of the other species within the Diaporthales, *Oc-j* is a canker pathogen, and is known to infect the hardwood butternut (*Juglans cinerea*) (Fig. 1). The Diaporthales order is composed of 13 families (*Voglmayr, Castlebury & Jaklitsch, 2017*), which include several plant pathogens, saprophytes, and enodphytes (*Rossman, Farr & Castlebury, 2007*). Numerous tree diseases are caused by members of this order. These diseases include dogwood anthracnose (*Discula destructiva*),

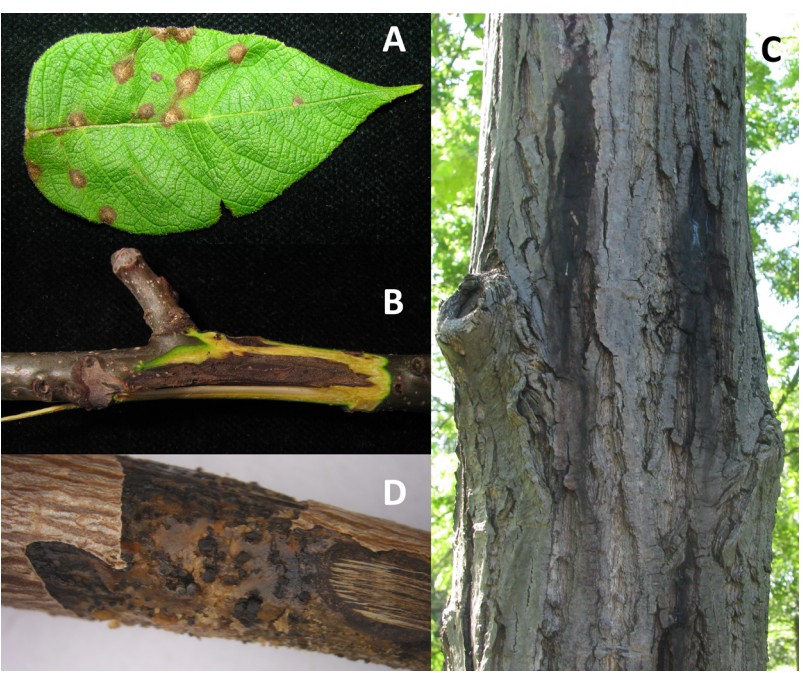

**Figure 1 Symptoms of infection caused by *Oc-j* on the (A) leaves, (B) branches and (C) trunk as well as as (D) signs of asexual fruting bodies on an infected branch.**

butternut canker (*Ophiognomonia clavigignenti-juglandacearum*), apple canker (*Valsa mali* and *Valsa pyri*), Eucalyptus canker (*Chrysoporthe autroafricana, C. cubensis* and *C. deuterocubensis*), and perhaps the most infamous and well-known chestnut blight (*Cryphonectira parastica*). Furthermore, several species in the Diaporthales also cause important disease of crops including soybean canker (*Diaporthe aspalathi*), soybean seed decay (*Diaporthe langicolla*) and sunflower stem canker (*Diaporthe helianthi*). In addition to pathogens, there are also a multitude of species that have a saprotrophic or endophytic life strategy (*Castlebury et al., 2002*). However, the saprophytic and endophytic species have not been studied as extensively as the pathogenic species in the Diaporthales.

Like many of the tree pathogens in Diaporthales, *Oc-j* is an invasive species, introduced into the U.S. from an unknown origin. This introduction caused extensive damage among the butternut population in North America during the latter half of the 20th century. The first report of butternut canker was in Wisconsin in 1967 (*Renlund, 1971*), and in 1979, the fungus was described for the first time as *Sirococcus clavigignenti-juglandacearum* (*Sc-j*) (*Nair, Kostichka & Kuntz, 1979*). Phylogenetic studies have determined the pathogen that causes butternut canker is a member of the genus *Ophiognomonia* and was reclassified as *Ophiognomonia clavigignenti-juglandacearum* (*Oc-j*) (*Broders & Boland, 2011*). The sudden emergence of *Oc-j*, its rapid spread in native North American butternuts, the scarcity of resistant trees, and low genetic variability in the fungus point to a recent introduction of a new pathogenic fungus that is causing a pandemic throughout North America (*Broders et al., 2012*).
While *Oc-j* is a devastating pathogen of the butternut, *Juglans cinerea*, there are also several species in the genus *Ophiognomonia* that are endophytes or saprophytes of tree species in the order *Fagales* and more specifically the *Juglandaceae* or walnut family (*Sogonov et al., 2008*; *Walker et al., 2012*). This relationship may support the hypothesis of a host jump, where the fungus may have previously been living as an endophyte or saprophyte before coming into contact with butternut. In fact, a recent study from China reported *Sirococcus* (*Ophiognomonia*) *clavigignenti-juglandacearum* as an endophyte of *Acer truncatum*, which is a maple species native to northern China (*Sun, Guo & Hyde, 2011*). The identification of the endophytic strain was made based on sequence similarity of the ITS region of the rDNA. A recent morphological and phylogenetic analysis of this isolate determined that while it is not *Oc-j*, this isolate is indeed more closely related to *Oc-j* than any other previously reported fungal species (*Broders et al., 2015*). The endophyic isolate also did not produce conidia in culture in comparison to *Oc-j* which produces abundant conidia in culture. It is more likely that these organisms share are common ancestor and represent distinct species.

While the impact of members of the Diaporthales on both agricultural and forested ecosystems is significant (*Rossman, Farr & Castlebury, 2007*), there has been limited information regarding the genomic evolution of this order of fungi. Several species have recently been sequenced and the genome data made public. This includes pathogens of trees and crops as well as an endophytic and saprotrophic species (*Yin et al., 2015*). However, these were generally brief genome reports and a more thorough comparative analysis of the species within the Diaporthales has yet to be completed.

Here we report the genome sequence of *Oc-j* and use it in comparative analyses with those of tree and crop pathogens within the Diaporthales. Comparative genomics of several members of the Diaporthales order provides valuable insights into fundamental questions regarding fungal lifestyles, evolution and phylogeny, and adaptation to diverse ecological niches, especially as they relate to plant pathogenicity and degradation of complex biomass associated with tree species.

## METHODS

### DNA extraction and library preparation

For this study, the ex-type culture of *Oc-j* (ATCC 36624) isolated from an infected butternut tree in Wisconsin in 1978, was sequenced. For DNA extraction, isolates were grown on cellophane-covered potato dextrose agar for 7–10 d, and mycelia was collected and lyophilized. DNA was extracted from lyophilized mycelia using the CTAB method as outlined by the Joint Genome Institute for whole genome sequencing (Kohler A, Francis M. Genomic DNA Extraction, Accessed 12/12/2015 http://1000.fungalgenomes.org/home/wp-content/uploads/2013/02/genomicDNAProtocol-AK0511.pdf). The total DNA quantity and quality were measured using both Nanodrop and Qubit, and the sample was sent to the Hubbard Center for Genome Studies at the University of New Hampshire, Durham, New Hampshire. DNA libraries were prepared using the paired-end Illumina Truseq sample preparation kit, and were sequenced on an Illumina HiSeq 2500.

## Genome assembly and annotation

We corrected our raw reads using BLESS 0.16 (*Heo et al., 2014*) with the following options: -kmerlength 23 -verify -notrim. Once our reads were corrected, we trimmed the reads at a phred score of 2 both at the leading and trailing ends of the reads using Trimmomatic 0.32 (*Bolger, Lohse & Usadel, 2014*). We used a sliding window of four bases that must average a phred score of 2 and the reads must maintain a minimum length of 25 bases. Next, de novo assembly was built using SPAdes-3.1.1 (*Bankevich et al., 2012*) with both paired and unpaired reads and the following settings: -t 8 -m 100 -only-assembler.

Genome sequences were deposited at ncbi.nlm.nih.gov under Bioproject number PRJNA434132. Gene annotation was performed using the MAKER2 pipeline (*Holt & Yandell, 2011*) in an iterative manner as is described in *Yu et al. (2016)*, with protein evidence from related species of *Melanconium* sp., *Cryphonectria parasitica*, *Diaporthe ampelina* (from jgi.doe.gov) and *Diaporthe helianthi* (*Baroncelli et al., 2016*), for a total of three iterations. PFAM domains were identified in all the genomes using hmmscan with trusted cutoff (*Bateman et al., 2004*). Only nine species were included in the PFAM analysis. For the three *Chrysoporthe* species, *D. aspalathi*, and *D. longicolla*, protein sequences were not readily available for downloading online and e-mail requests were unsuccessful. Secondary metabolism gene clusters were identified using antiSMASH 4.0 (*Weber et al., 2015*).

## Species phylogeny

Core eukaryotic proteins identified by CEGMA (*Parra, Bradnam & Korf, 2007*) were first aligned by MAFFT (*Katoh et al., 2002*) and then concatenated. Only proteins that were present in all genomes and all sequences were longer than 90% of the *Saccharomyces cerevisiae* ortholog were used. This resulted in a total of 340 single-copy genes that were concatenated into a single alignment. Phylogeny was then inferred using maximum likelihood by RAxML (*Stamatakis, 2014*) with PROTGAMMA used to estimate protein evolution, and with 100 bootstraps and midpoint rooted.

## Secreted CAZyme prediction

SignalP (*Petersen et al., 2011*) was used to predict the presence of secretory signal peptides. CAZymes were predicted using CAZymes Analysis Toolkit (*Park et al., 2010*) based on the most recent CAZY database (www.cazy.org). Proteins that both contain a signal peptide and are predicted to be a CAZyme are annotated as a secreted CAZyme.

# RESULTS AND DISCUSSION

## Genome assembly and annotation of *Oc-j*

The draft genome assembly of *Oc-j* contains a total of 52.6 Mb and 5,401 contigs, with an N50 of 151 Kbp (Table 1). The completeness of the genome assembly was assessed by identifying universal single-copy orthologs using BUSCO with lineage dataset for Sordariomycota (*Simão et al., 2015*). Out of 3,725 total BUSCO groups searched, we found that 3,378 (90.7%) were complete and 264 (7.1%) were fragmented in the *Oc-j* genome, while only 83 (2.2%) were missing. This result indicates that the *Oc-j* genome is relatively complete.

**Table 1** Assembly statistics of the butternut canker pathogen and other Diaporthales analyzed.

| Species | Disease | Assembly size (Mb) | Scaffolds | Scaffold N50 lengh (Kb) | Scaffold L50 count | Total genes | References |
|---|---|---|---|---|---|---|---|
| *Ophiognomonia clavigignenti juglandacearum* | Butternut canker | 52.6 | 5,401 | 151 | 104 | 11,247 | This study |
| *Valsa pyri* | Apple Canker | 35.7 | 475 | 239 | 43 | 10,481 | *Yin et al. (2015)* |
| *Valsa mali* | Apple Canker | 44.7 | 353 | 3 | 6 | 11,261 | *Yin et al. (2015)* |
| *Cryphonectria parasitica* | Chestnut blight | 43.9 | 26 | 5.1 | 4 | 11,609 | jgi.doe.gov |
| *Chrysoporthe cubensis* | Eucalyptus Canker | 42.6 | 954 | 156 | 85 | 13,121 | *Wingfield et al. (2015a)* |
| *Chrysoporthe deuterocubensis* | Eucalyptus Canker | 44.0 | 2,574 | 84 | 147 | 13,772 | *Wingfield et al. (2015a)* |
| *Chrysoporthe austroafricana* | Eucalyptus Canker | 44.7 | 6,415 | 47 | 264 | 13,484 | *Wingfield et al. (2015b)* |
| *Diaporthe helianthi* | Sunflower Stem Canker | 63.7 | 7,376 | 20 | 860 | 14,220 | *Baroncelli et al. (2016)* |
| *Diaporthe ampelina* | Endophyte | 59.5 | 904 | 135 | 123 | 24,672 | *Savitha, Bhargavi & Praveen (2016)* |
| *Diaporthe aspalathi* | Soybean Canker | 55.0 | 1,871 | 87 | 184 | 14,962 | *Li et al. (2016)* |
| *Diaporthe longicolla* | Soybean Seed Decay | 64.7 | 985 | 204 | 86 | 15,738 | *Li et al. (2015)* |
| *Melanconium spp.* | Endophyte | 58.5 | 100 | 1.94 | 9 | 16,656 | jgi.doe.gov |
| *Magnaporthe grisea* | Rice Blast | 41.7 | 30 | 2.89 | 6 | 11,109 | *Dean et al. (2005)* |
| *Neurospora crassa* | bread mold | 41.0 | 20 | 6 | 3 | 10,785 | *Galagan et al. (2003)* |

## Genome-wide multi-gene phylogeny of Diaporthales

A total of 340 genes were used to generate a multigene phylogeny of the order Diaporthales. All sequenced species in Diaporthales (one genome per species for 12 species) were used. These species include: *Cryphonectria parasitica* (jgi.doe.gov), *Chrysoporthe cubensis* (*Wingfield et al., 2015b*), *Chrysoporthe deuterocubensis* (*Wingfield et al., 2015b*), *Chrysoporthe austroafricana* (*Wingfield et al., 2015a*), *Valsa mali* (*Yin et al., 2015*), *Valsa pyri* (*Yin et al., 2015*), *Diaporthe ampelina* (*Savitha, Bhargavi & Praveen, 2016*), *Diaporthe aspalathi* (*Li et al., 2016*), *Diaporthe longicolla* (*Li et al., 2015*), *Diaporthe helianthi* (*Baroncelli et al., 2016*), and *Melanconium* sp. The outgroups used were *Neurospora crassa* (*Galagan et al., 2003*) and *Magnaporthe grisea* (*Dean et al., 2005*). We show that the order Diaporthales is divided into two main branches. One branch includes *Oc-j*, *C. parasitica*, and the *Chrysoporthe* species (Fig. 2), in which *Oc-j* is located basally, indicating its early divergence from the rest of the branch. The other branch includes the *Valsa* and *Diaporthe* species.

## Gene content comparison across the Diaporthales

To examine the functional capacity of the *Oc-j* gene repertoire, the PFAM domains were identified in the protein sequences (Table S1). For comparison, we also examined eight of

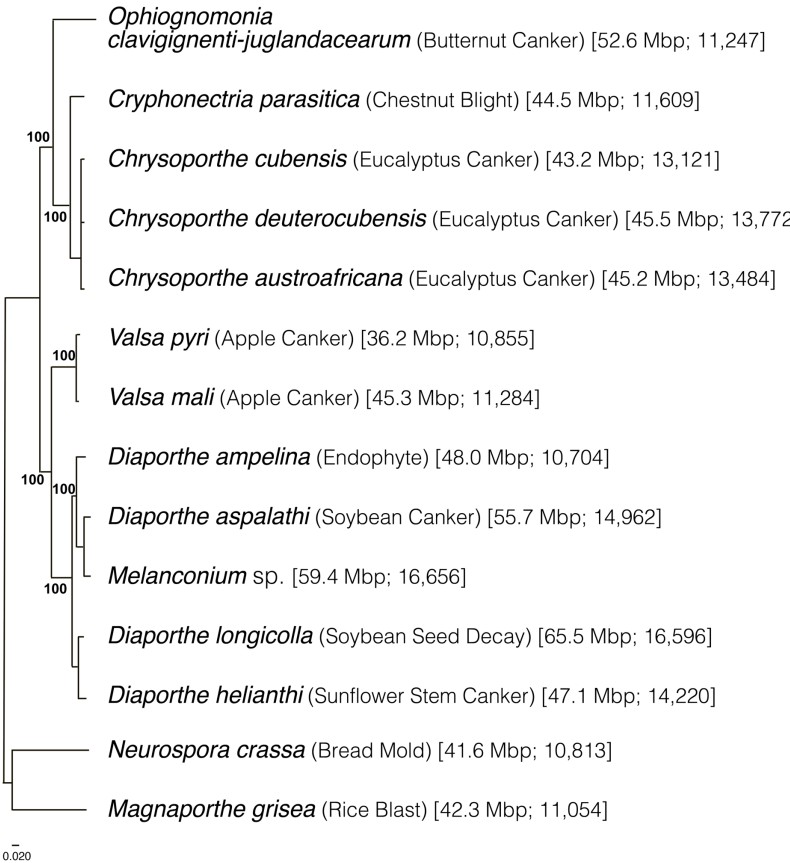

**Figure 2 Phylogeny of Diaporthales and related species inferred using maximum likelihood by RAxML (*Yu et al., 2016*) with 1,000 bootstraps and then midpoint rooted.** The first and second numbers in parentheses represent the genome sizes in Mb and the number of predicted protein models, respectively.               

the above mentioned 13 related species, where protein sequences were successfully retrieved (see "Methods" for details).

CAZymes are a group of proteins that are involved in degrading, modifying, or creating glycosidic bonds and contain predicted catalytic and carbohydrate-binding domains (*Morales-Cruz et al., 2015*). When secreted by fungal pathogens, CAZymes can participate in degrading plant cell walls during colonization; therefore, a combination of CAZyme and protein secretion prediction was used to identify and classify enzymes likely involved in cell wall degradation in plant pathogens (*Morales-Cruz et al., 2015*). Overall, the *Oc-j* genome contains 576 putatively secreted CAZymes, more than any of the other eight species included in this analysis, and 60 CAZymes more than *D. helianthi*, the species with the second most (Fig. 3; Table S1). Given that all except the bread mold species *N. crassa*, which has the lowest number of secreted CAZymes, are plant pathogens, this result indicates that *Oc-j* contains an especially large gene repertoire for cell wall degradation. However, it is not the greatest among members of the kingdom fungi. For instance, the saprophytic species *Penicillium subrubescens* and *Podospora anserina* have been found to have 719 and 590 CAZymes, respectively (*Peng et al., 2017*). This pattern also holds true
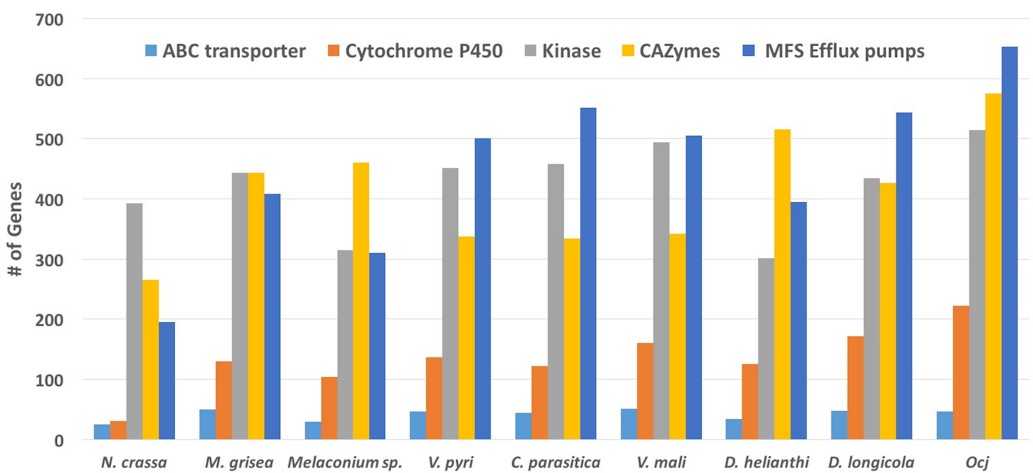

**Figure 3** Abundance of genes in specific classes including ABC transporters, Cytochrome P450, Kinases, CAZymes, and MFS Efflux pumps present in nine species within the order Diaporthales.

when compared to other tree pathogens that have a latent period and an extended host range. For instance, *Botryosphaeria dothidea*, which is a pathogen of many woody species and has a prolonged latent and endophytic phase during the infection process, was reported to have 623 CAZymes (*Marsberg et al., 2017*). Other studies have also found an elevated number of CAZymes in multi-host pathogens of woody plant species including *Eutypa lata* and *Neofusicoccum parvum* which had 484 and 413 CAZymes, respectively (*Morales-Cruz et al., 2015*). Since secreted CAZymes are involved in degrading plant cell walls, the large number of secreted CAZymes in *Oc-j* and other pathogens of woody plants might help facilitate a life-style infecting multiple species of perennial woody plants with a wide array of both preformed and infection-induced resistance responses.

A total of 77 kinase related PFAM domains found in the nine above mentioned species. Of these, 61 domains are more abundant in *Oc-j* than the average of the eight related species, while only eight domains are less abundant in *Oc-j* (Fig. 3; Table S1). Given that kinases are involved in signaling networks, this result might indicate a more complicated signaling network in *Oc-j*. One example of kinase gene family expansion is the domain family of fructosamine kinase (PF03881). *Oc-j* contains 29 genes with this domain, while *N. crassa* has none and the other related species have between 3 and 8 genes (Fig. 3). Almost nothing is known about the function of fructosamine kinases in fungi. In other eukaryotes, it is involved in protein deglycation by mediating phosphorylation of fructoselysine residues on glycated proteins. While plants and fungi both produce an array of glycated proteins it is unclear if *Oc-j* is producing fructosamine kinase to function in deglycation of proteins produced by the plant host. Fructosamine kinases are extracellular serine-proteases, and are implicated in the pathogenic activity of several fungal pathogens of mammals (*Da Silva et al., 2006*), including *Aspergillus fumigatus* (*Monod et al., 2002*). Whether or not fructosamine kinase functions are important for plant pathogens remains unknown.

Cytochrome P450s (CYPs) are a superfamily of monooxygenases that play a wide range of roles in metabolism and adaptation to ecological niches in fungi (*Chen et al., 2014*). Due to their participation in a large number of detoxification reactions as well as in the metabolism of specific xenobiotics which may be co-assimilated as carbon source, CYPs are thought to be critical for the colonization of new ecological niches (*Moktali et al., 2012*). Among the nine species included in the PFAM analysis, *N. crassa* has only 31 CYPs, while the other eight plant-pathogenic species have between 104 and 223 CYPs. This result likely reflects that the ecological niche of *N. crassa*, saprphyte, is more favorable for fungal growth than alive plants with active defense mechanisms, thus fewer CYPs are needed for *N. crassa* to cope with relatively simple substrates. The *Oc-j* genome contains 223 CYPs (Fig. 3; Table S1), more than any of the other seven plant-pathogenic fungal species. This may be an evolutionary response to the number and diversity of secondary metabolites, such as juglone, produced by butternut as well as other *Juglans* species, that would need to be metabolized by any fungal pathogen trying to colonize the tree. Other pathogens of woody plants species are also known to have an increased number of CYPs. The grapevine canker pathogens *E. lata* and *N. parvum* have 205 and 212 CYPs respectively (*Morales-Cruz et al., 2015*). According to the Fungal Cytochrome P450 Database (FCPD) (*Nelson, 2009*), very few fungal species have more CYPs than *Oc-j*. The brown rot and pathogenic fungi in the basidiomycota have the greatest number of CYPs in the fungal kingdom. The brown rot fungus *Postia placenta* has the greatest number of CYPS with 345 CYPs, while the cacao pathogen *Moniliophthora perniciosa* and root rot pathogen *Armillaria mellea* have 300 and 245 CYPs, respectively (*Nelson, 2009*). The one characteristic all of these fungi have in common is their association with dead, dying or living trees and the need to breakdown or resist the multitude of chemical compounds produced by tree species.

To overcome host defenses, infect and maintain colonization of the host, fungi employ efflux pumps to counter intercellular toxin accumulation (*Coleman & Mylonakis, 2009*). Here, we examine the presence of two major efflux pump families, the ATP-binding cassette (ABC) transporters and transporters of the major facilitator superfamily (MFS) (*Coleman & Mylonakis, 2009*) in the Diaporthales genomes. ABC transporters (PF00005, pfam.xfam.org) are present in all nine species included in the PFAM analysis, ranging from 25 genes in *N. crassa* to 51 genes in *V. mali*, while *Oc-j* has 47 genes (Fig. 3; Table S1). The major facilitator superfamily are membrane proteins are expressed ubiquitously in all kingdoms of life for the import or export of target substrates. The MFS is a clan that contains 24 PFAM domain families (pfam.xfam.org). The *Oc-j* genome contains 653 MFS efflux pump genes, the most among all nine species included in the analysis (Fig. 3; Table S1). *Cryphonectria parasitica* has 552 genes, the second most. Interestingly, it was recently shown that the secondary metabolite juglone extracted from *Juglans* spp. could be used as potential efflux pump inhibitors in *Staphylococcus aureus*, inhibiting the export of antibiotics out of the bacterial cells (*Zmantar et al., 2016*). In line with this previous finding, our results suggest that *Oc-j* genome contains a large arsenal of efflux pumps likely

due to the need to cope with hostile secondary metabolites such as the efflux pump inhibitor juglone.

*Secondary metabolism gene clusters.* Secondary metabolite toxins play an important role in fungal nutrition and virulence (*Howlett, 2006*). To identify gene clusters involved in the biosynthesis of secondary metabolites in *Oc-j* and related species, we scanned their genomes for such gene clusters. We found that the *Oc-j* genome contains a remarkably large repertoire of secondary metabolism gene clusters, when compared to the closely related *C. parasitica* and the *Chrysoporthe* species (Fig. 3; Table S1). While *C. parasitica* has 44 gene clusters and the *Chrysoporthe* species have 18–48 gene clusters, the *Oc-j* genome has a total of 72 gene clusters (Table S1), reflecting a greater capacity to produce various secondary metabolites. Among the 72 gene clusters in *Oc-j*, more than half are type 1 polyketide synthases (t1pks, 39 total, including hybrids), followed by non-ribosomal peptide synthetases (nrps, 14 total, including hybrids) and terpene synthases (9) (Table S1).

When compared to all species included in this study, we found that *Oc-j* has the third highest number of clusters, only after *D. longicolla* and *Melanconium* sp.; however, when normalized by total gene number in each species, *Oc-j* is shown to have 6.4 secondary metabolism gene clusters per 1,000 genes, the highest among all species included in this study. The *D. helianthi* genome contains only six clusters, likely due to the poor quality of the genome assembly with an N50 of ~6 kb (Fig. 3; Table S1). Cluster numbers can vary greatly within the same genus.

## CONCLUSION

We constructed a high-quality genome assembly for *Oc-j*, and delineated the phylogeny of the Diaporthales with a genome-wide multi-gene approach, revealing two major branches. We then examined several gene families relevant to plant pathogenicity and complex biomass degradation. We found that the *Oc-j* genome contains large numbers of genes in these gene families. These genes might be essential for *Oc-j* to cope with its niche in the hardwood butternut (*Juglans cinerea*). Future research will need to focus on understanding the prevalence of these genes associated with complex biomass degradation among other members of the *Ophiognomonia* genus which include endophytes, saprophytes and pathogens. It will be interesting to know which of these different classes of genes are important in the evolution of distinct fungal lifestyles and niche adaptation. Future research will also focus on comparing the butternut canker pathogen to canker pathogens of other tree species which produce large numbers of secondary metabolites known to be important in host defense. Our genome also serves as an essential resource for the *Oc-j* research community.

## ACKNOWLEDGEMENTS

Special thanks to Dr. Matt MacManus. This started as an in-class project for PhD student TS and provided the basis for this final manuscript.

### Funding

Funding from this project was provided from grant number 14NB49 from the National Geographic Society. Dr. Broders was also supported by the Simon's Foundation Grant number 429440 to the Smithsonian Tropical Research Institute. The funders had no role in study design, data collection and analysis, decision to publish, or preparation of the manuscript.

### Grant Disclosures

The following grant information was disclosed by the authors:
National Geographic Society: 14NB49.
Simon's Foundation: 429440.

### Competing Interests

The authors declare that they have no competing interests.

### Author Contributions

- Guangxi Wu analyzed the data, prepared figures and/or tables, authored or reviewed drafts of the paper, and approved the final draft.
- Taruna A. Schuelke conceived and designed the experiments, performed the experiments, analyzed the data, prepared figures and/or tables, authored or reviewed drafts of the paper, and approved the final draft.
- Gloria Iriarte conceived and designed the experiments, performed the experiments, authored or reviewed drafts of the paper, and approved the final draft.
- Kirk Broders conceived and designed the experiments, authored or reviewed drafts of the paper, and approved the final draft.

### Data Availability

The data are available in NCBI: PRJNA434132.

### Supplemental Information

Supplemental information for this article can be found online at http://dx.doi.org/10.7717/peerj.9265#supplemental-information.

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
