# Peer review of "The genome of the butternut canker pathogen, Ophiognomonia clavigignenti-juglandacearum shows an elevated number of genes associated with secondary metabolism and protection from host resistance responses"

_PeerJ, doi:10.7717/peerj.9265_

## Round 0.1 · original submission · Major Revisions

The reviewers' comments were positive, but they also indicated a need for more detailed explanations of methods, some of the results, and potential discrepancies in gene numbers among genomes and publications.

Reviewer 1 ·

Basic reporting

The report and the research are well developed and follow the requirements for clarity, background, data sharing and related. Below some minor comments.
- There are several typos in the scientific names of fungal species and in some other few places. In the corrected file I have attached you can find where this needs revision.
- It is a good idea to check that the names of fungal species are the current ones in use. Names of fungi change frequently so I do this exercise pretty frequently. While most of the names reported by the authors are correct, the current name of one of them, Diaporthe longicolla should be changed to Phomopsis longicolla, wherever is needed. You can consult Mycobank.org or Indexfungorum.org for this. This correction can also be found in the attached file.
- Lines 61-63 need some attention regaarding the statements made. Find the comments in the attached file.

Experimental design

- This is the first report for the pathogen in questions, which fits into the scope of original and primary research.
- Regarding the phylogenetic analyses. Needs a little more information, for example how the authors obtained the core eukaryotic proteins. Orthology? I have seen in other studies only the orthologous genes that are in single copy in all individuals are used for the phylogenetic analysis....was this the same in this case? Also, what was the final length of the alignment? did you use any model of protein evolution?

Validity of the findings

No comments. The authors conclusions are supported by the data they obtained.

Additional comments

It was a very nice, simple work to read.

Annotated reviews are not available for download in order to protect the identity of reviewers who chose to remain anonymous.

·

Basic reporting

As indicated in my Comment #1, I suggest that thef manuscript incude a table summarizing structural and functional genome statistics. Some data (on CYP450 gene annotations) seem to be missing from the Supplementary Table provided by the authors.

Experimental design

As indicated in my Comment #2, some of the genomic data provided by the authors (for genomes sequenced by other investigators) can be quite different from data available in public databases or in previous publications. I believe this should be clarified.

Validity of the findings

As indicated in my Comment #3, I feel the discussion is a bit weak and could be strengthened by making a better use of the existing fungal genomics literature.

Additional comments

The work reported by Wu and colleagues, based on the sequencing and comparative analysis of the butternut canker pathogen Ophiognomonia clavigignenti-juglandacearum (Oc-j), represents an interesting and potentially very useful contribution to the study of fungal ganets of cankers. The manuscript is generally well written, concise and easy to read. In my opinion, however, the manuscript would be improved by providing additional information and addressing additional issues. In particular :

1. In the opening section of Results and Discussion (Genome assembly and annotation of Oc-j), the authors should provide more data, including a table summarizing structural and functional genome statistics (including total number of genes identified per genome) for Oc-j and other Diaporthales genomes they analyzed. The estimated size of the Oc-j genome (52.6 Mb) seems quite high given the number of predicted genes (11 247) and I would expect the authors to discuss this issue (assembly artefact or biological reality?).

2. I was surprised by some of the genomic data provided by the authors. For example, on L118, they indicate that the genome of Grosmannia clavigera contains 530 CAZyme-encoding genes, which is far superior to the number (n=269) reported by DiGuistini et al (2011) who sequenced and annotated the G. clavigera genome. Some of the data from non-Diaporthales species originate from a previous paper (Schuelke et al. 2017) but, when I checked this reference, it was not clear to me how Schuelke and colleagues had obtained gene numbers that differed so much from those reported elsewhere. Another example : in Supplementary Table «Antismash », the authors indicate that the Neurospora crassa contains 10 813 genes, which again is much higher than numbers usually reoprted for this species (e.g. 9 758 coding genes according to Ensembl Fungi). The reader is referred to Suppl. Table 1 for data on CYP genes but, unless I missed something, no relevant information is provided here. I would therefore like the authors to provide background information and further precisions on this issue of genomic data.

3. I would strongly encourage the authors to give further thought to the discussion of their results. While I understand they wanted to focus on the Diaporthales, I feel this weakens several of their interpretations. In the case of genes encoding secreted CAZymes (105-121), the authors’ suggestion that «the large number of secreted CAZymes in Oc-j might help facilitate its life-style infecting and colonizing butternut trees » is based on a limited dataset (this work and the previous work by Schuelke et al. [2017 ]). Furthermore, the authors only consider the breadmold Neurospora crassa when discussing saprotrophs. The genomes of several fungal saprotrophs (e.g. Chaetomium globosum, Podospora anserina) contain high numbers of CAZy-encoding genes, and this would seem relevant in the discussion.
When discussing kinases (L122-129), the authors mention a potentially interesting element (gene family expansion in the domain family of fructosamine kinase PF03881 but do not discuss this finding. The number of CYP-encoding genes annotated in Oc-j (n = 223) is indeed very high compared with numbers estimated for other species considered in the study. I think it would be interesting to put this in a wider perspective by comparing with data from other studies and discussing further the families and potential functions represented in the OC-j CYP gene dataset.

Minor points

L35. Typo. Should read "endophytes"

L60-64. Two different works published 5 years apart are each presented as «a recent study». I suggest changing the wording.

L122 and L130 : Suggestion : Use the plural form for « Kinase » and « Cytochrome P450 ».

Legend of Figure 1: Suggestion: indicate whether Figure 1D shows asexual or sexual fruiting bodies of Oc-j.

Figure 2 : Not a bad idea to include genome size and gene number data in the figure but I think it would help the reader if this oinformation (and additoinal genomic information) was also provided in a Table within the manuscript.

Legend of Figure 2. Typo in last sentence. Should read: "...number of predicted..."

---

## Round 0.2 · accepted · Accept

The reviewer found some typos that you may want to fix before publication. See the annotated manuscript they provided.

·

Basic reporting

No comment

Experimental design

No comment

Validity of the findings

No comment

Additional comments

I have read the revised manuscript submitted by the authors and I feel thay have addressed concerns expressed by both reviewers of the original manuscript. In particular, I find the inclusion of Table 1 (Assembly statistics) a useful addition and I believe the conclusions are now presented in a more nuanced way. I am also happy with the authors' detailed response to reviewers' comments.

I have attached an annotated copy of the revised manuscript, with only a few very minor corrections (in red) and comments (mostly related to typos). I suggest adding one reference when discussing protein deglycation by fructosamine kinases in eukaryotes other than fungi (L181-182).